# A Thermotolerant Yeast *Cyberlindnera rhodanensis* DK Isolated from Laphet-so Capable of Extracellular Thermostable β-Glucosidase Production

**DOI:** 10.3390/jof10040243

**Published:** 2024-03-23

**Authors:** Nang Nwet Noon Kham, Somsay Phovisay, Kridsada Unban, Apinun Kanpiengjai, Chalermpong Saenjum, Saisamorn Lumyong, Kalidas Shetty, Chartchai Khanongnuch

**Affiliations:** 1Division of Biotechnology, School of Agro-Industry, Faculty of Agro-Industry, Chiang Mai University, Chiang Mai 50100, Thailand; nwenoonkham@gmail.com (N.N.N.K.); somsay2009@hotmail.com (S.P.); 2Division of Food Science and Technology, School of Agro-Industry, Faculty of Agro-Industry, Chiang Mai University, Chiang Mai 50100, Thailand; kridsada.u@cmu.ac.th; 3Division of Biochemistry and Biochemical Innovation, Department of Chemistry, Faculty of Science, Chiang Mai University, Chiang Mai 50200, Thailand; apinun.k@cmu.ac.th; 4Department of Pharmaceutical Science, Faculty of Pharmacy, Chiang Mai University, Chiang Mai 50200, Thailand; chalermpong.s@cmu.ac.th; 5Department of Biology, Faculty of Science, Chiang Mai University, Chiang Mai 50200, Thailand; saisamorn.l@cmu.ac.th; 6Global Institute of Food Security and International Agriculture (GIFSIA), Department of Plant Sciences, North Dakota State University, Fargo, ND 58108, USA; kalidas.shetty@ndsu.edu; 7Research Center for Multidisciplinary Approaches to Miang, Multidisciplinary Research Institute, Chiang Mai University, Chiang Mai 50200, Thailand; 8Research Center for Microbial Diversity and Sustainable Utilization, Chiang Mai University, Chiang Mai 50200, Thailand

**Keywords:** Laphet, thermotolerant yeast, β-glucosidase, carbon sources, Plackett–Burman design, *Cyberlindnera rhodanensis*

## Abstract

This study aims to utilize the microbial resources found within Laphet-so, a traditional fermented tea in Myanmar. A total of 18 isolates of thermotolerant yeasts were obtained from eight samples of Laphet-so collected from southern Shan state, Myanmar. All isolates demonstrated the tannin tolerance, and six isolates were resistant to 5% (*w*/*v*) tannin concentration. All 18 isolates were capable of carboxy-methyl cellulose (CMC) degrading, but only the isolate DK showed ethanol production at 45 °C noticed by gas formation. This ethanol producing yeast was identified to be *Cyberlindnera rhodanensis* based on the sequence analysis of the D1/D2 domain on rRNA gene. *C. rhodanensis* DK produced 1.70 ± 0.01 U of thermostable extracellular β-glucosidase when cultured at 37 °C for 24 h using 0.5% (*w*/*v*) CMC as a carbon source. The best two carbon sources for extracellular β-glucosidase production were found to be either xylose or xylan, with β-glucosidase activity of 3.07–3.08 U/mL when the yeast was cultivated in the yeast malt extract (YM) broth containing either 1% (*w*/*v*) xylose or xylan as a sole carbon source at 37 °C for 48 h. The optimal medium compositions for enzyme production predicted by Plackett–Burman design and central composite design (CCD) was composed of yeast extract 5.83 g/L, peptone 10.81 g/L and xylose 20.20 g/L, resulting in a production of 7.96 U/mL, while the medium composed (g/L) of yeast extract 5.79, peptone 13.68 and xylan 20.16 gave 9.45 ± 0.03 U/mL for 48 h cultivation at 37 °C. Crude β-glucosidase exhibited a remarkable stability of 100%, 88% and 75% stable for 3 h at 35, 45 and 55 °C, respectively.

## 1. Introduction

Laphet or Laphet-so, a traditional fermented tea of Myanmar, is prepared by fermentation of young tea leaves, which are commonly found all over the mountainous regions of Myanmar, especially the cultivated tea garden in Shan state [1]. Laphet is predominantly prepared in an edible form and served as a dessert [2]. The benefits and microbial involvement in Laphet or tea fermentation, as well as the presence of living microorganisms within Laphet, contribute to its high content of total polyphenols and total flavonoids, resulting in antioxidant activity comparable to that of green tea and black tea sourced from various locations in Myanmar [3]. The microbial diversity within Laphet was investigated, and it was found that the predominant bacterial genera identified in Laphet were *Lactobacillus*, *Acetobacter*, *Pediococcus*, *Lactobacillaceae*, *Enterobacteriaceae* and *Pseudomonas*, while *Saccharomycetales*, *Debaryomyces, Dipodascus*, *Cyberlindnera*, *Pichia* and *Candida* were found as the dominant yeast genera in Laphet-so products [4]. However, investigations on individual microorganisms isolated from Laphet have so far lacked comprehensive information regarding their specific biological communities and their roles in the fermentation of tea leaves.

According to the previous study on microbial diversity in Laphet by Bo et al. [4], the important microbes found in Laphet are yeast, lactic acid bacteria and acetic acid bacteria, which corresponds to a previous report describing microorganisms involved in fermentation of Miang, a traditional fermented tea from north Thailand, which stated that the most significant groups of microbes involved in Miang fermentation are lactic acid bacteria, yeast and endospore forming *Bacillus* spp. [5,6,7,8]. Kodchasee et al. [9] also described the importance of a filamentous fungus *Aspergillus niger* on Miang fermented by a filamentous fungi growth-based fermentation process (FFP). The microbial resources in Miang samples showed the unique characteristics such as tannin tolerance, nutritional biotransformation, functional metabolites and thermotolerant properties. Some microbes showed the potential to be used in a variety of applications [7,10,11]. Regarding the typical process of the traditional preparation method for Laphet-so fermentation, the harvested tea shoots have to be processed by steaming or exposing to heat at 95–100 °C for 10–20 min, followed by rolling, packing them into plastic bags or clay pots, then pressing the leaves with heavy weights. Natural fermentation was carried out for two weeks to a year without the addition of external microbial inoculation. This process results in color changes, softened texture and flavor development. Similar to Miang, Laphet is rich in phenolic compounds such as tannins and exhibits microbial diversity [4,12]. Therefore, a Laphet sample is expected to be a source of diverse microbes particularly tolerant of high tannins and high temperature as those have passed and withstand the heat treatment process during the typical traditional fermentation process. The thermotolerance of specific yeast is of significant interest in several research fields, including bioethanol production and thermal food processing [13]. Furthermore, thermotolerant yeast strains are revealed to possess β-glucosidase production ability through the hydrolysis of cellulose to yield fermented sugar, a procedure known as saccharification [14]. In addition, thermotolerant yeast, capable of producing both β-glucosidase and ethanol, is considered a promising candidate not only for its efficient conversion of cellulose to ethanol but also as a valuable model for exploring the conformational stability of large isozymic proteins. This is particularly significant due to their involvement in a wide array of crucial biological processes, including but not limited to growth, development, chemical defense, interactions between hosts and parasites and cell signaling [14,15]. The thermotolerant yeast strains identified as *Periconia* sp. and *Pichia* sp. with a capability of high thermostable β-glucosidase production have been also reported [16].

β-Glucosidase, also systematically known as β-D-glucoside glucohydrolase (EC 3.2.1.21), is synthesized by a wide range of organisms [17]. The enzymes from microbes, plants and animals play essential roles in various biological processes and also have been utilized in many industrial processes; however, microbial enzymes are favored due to their vast diversity of enzymes with a wide range of specificities and stability under varying pH, temperature and other environmental conditions that are more suitable for industrial processes. Furthermore, they can be genetically engineered to enhance productivity and specific enzyme characteristics and be free from contaminants, making them cost-effective to produce on a large scale compared to extracting enzymes from plants or animals by the fermentation process [18]. The role of β-glucosidase is crucial for alleviating the inhibition caused by cellobiose by converting it into glucose and/or cleaving glycosidic bonds, facilitating the breakdown of glycosides into their aglycones and sugars from glycosidically bound non-volatiles and volatiles compounds [19,20]. In addition, β-glucosidase is capable not only of incorporating in cellulose hydrolysis also plays an important role in liberating aroma compounds present in fruits and fermenting products, as highlighted by Singh et al. [21]. The considerable research on enzymes like β-glucosidase found in yeast *Candida* sp. have been extensively studied for their capacity to release aromatic compounds from their inert glucosidic precursors [22]. This enzymatic hydrolysis in the content of tea products has led to significant improvements in the aroma of such products [23]. Furthermore, Wang et al. [24] revealed that the β-glucosidase production in *Cyberlindnera* (Williopsis) *saturnus* var. *mrakii* resulted in the generation of a diverse array of tea aroma compounds. Some previous reports described the attempts to increase β-glucosidase production using various strategies including optimization [25], microbial co-culture [26], genetic modification using recombinant DNA technology [27] and heterologous expression [28]. The duel character (anabolic and catabolic) of β-glucosidases allows for their further applications not only in improving the aroma and flavor of beverages such as wine, soy drink and tea but also in further improving fragrances and cosmetic formulations as well as other applications in the chemical, pharmaceutical and detergent industries [21].

*C. rhodanensis* stands out in microbiology due to its diverse ecological and industrial significance. Notably, it plays a pivotal role in ethanol fermentation under thermotolerant conditions, as described by Keo-oudone et al. [29]. Beyond its prowess in ethanol production, specific strains of *C. rhodanensis*, namely A22.3 and A45.3, have been identified as producers of tannase, an enzyme previously isolated from Miang samples [30]. This newfound capability opens up promising avenues for industrial applications, given the versatile uses of tannase in sectors such as food processing and pharmaceuticals. Expanding the canvas of its industrial importance, Kodchasee et al. [10] shed light on an intriguing aspect of *C. rhodanensis*: its dual proficiency in both synthesizing the β-glucosidase enzyme and demonstrating a unique ability to produce esters. These esters, essential components in the intricate process of wine production, unveil an additional layer of potential applications for *C. rhodanensis*, especially within the wine industry. While previous recognition has centered on its β-glucosidase synthesis, the newfound capacity for ester production enhances the microorganism’s versatility and underscores its potential significance in various industrial processes. In the case of Laphet, it is worth noting that microorganisms producing β-glucosidase could potentially play a crucial role in shaping the unique flavors and characteristics of tea during the fermentation process, as discussed by Zhang et al. [31]. However, research in this area is still in its infancy, and the isolation and characterization of such yeast strains from Laphet-so hold promise not only for understanding the intricacies of temperature-sensitive transformation but also for their industrial application.

This study describes the isolation and characterization of thermotolerant yeast strains from Laphet, shedding light on their metabolic and enzyme-producing capabilities, with a particular focus on their potential for β-glucosidase production. The nutritional components optimization predicted for higher enzyme production by a statistic experimental designs model is also described.

## 2. Materials and Methods

### 2.1. Chemicals and Microbial Strains

All chemicals utilized for preparing solvents, reagents and buffers were of analytical grade. The analytical grade of *p*-nitrophenyl-β-D-glucopyranoside, *p*-nitrophenyl-β-D-mannopyranoside and *p*-nitrophenyl-β-D-xylopyranoside were purchased from Sigma-Aldrich (St. Louis, MO, USA). The reference strain *Cyberlindnera rhodanensis* TBRC-BCC 64435 was obtained from Thailand Bioresource Research Center (Bangkok, Thailand). *C. rhodanensis* A.45.3 previously isolated from Miang samples [30] was also used as the second reference strain.

### 2.2. Laphet-so Sampling Sites and Sample Collection

Laphet-so samples were collected from eight sampling sites located in the southern Shan state of Myanmar, consisting of Mong Hsu, Mong Hsu Township Namphet village, Laihka, Pang Long, Kunhing, Taunggyi, Pindaya Township Myaadai village and Tanngyo village. Laphet-so samples were obtained from local Laphet producers and local markets.

### 2.3. Isolation of Thermotolerant Yeasts

Each Laphet-so sample (50 g) was mixed thoroughly with 200 mL of sterilized 0.85% (*w*/*v*) NaCl, serving as a diluent. The homogenization process was carried out by using a Masticator homogenizer (IUL Instruments, Barcelona, Spain) for a uniform suspension for 5 min. Five mL of each homogenate suspension was subjected to heat treatment at 80 °C for 40 min and serially diluted to 10^−1^–10^−5^; 0.1 mL of each heat-treated suspension was spread on yeast malt extract (YM) agar (3 g/L yeast extract, 3 g/L malt extract, 5 g/L peptone, 10 g/L glucose and 20 g/L agar), supplemented with 100 mg/L chloramphenicol and incubated at 45 °C for 3 days. The single colony of distinguished yeast colony was picked up and stored at −20 °C in YM broth consisting of 50% (*v*/*v*) glycerol [7].

### 2.4. Characterization of Thermotolerant Yeasts

#### 2.4.1. Evaluation of Tannin-Tolerant Ability

To assess the tannin-tolerant ability of yeast isolates, a single colony from each yeast isolate was selected and streaked on yeast malt extract (YM) agar media supplemented with varying concentrations of tannic acid. Subsequently, the growth of yeast isolates was monitored after incubating the culture plates at 45 °C for 3 days. Beforehand, yeast malt extract (YM) agar was prepared separately by adding different concentrations of tannic acid solution, and the mixture was then autoclaved at 121 °C for 20 min. Sterile tannic acid solutions were aseptically pipetted into the melted yeast malt extract (YM) agar to achieve final concentrations of 0.5%, 1%, 3% and 5% (*w*/*v*) of tannic acid before being utilized for the experiments [7].

#### 2.4.2. Ethanol Producing Capability

The ability of ethanol production at high temperatures (45 °C) was assessed by gas production in a Durham tube. A single colony of each yeast isolate was picked up with a sterile toothpick and inoculated in a test tube containing 10 mL of yeast malt extract (YM) broth with 2% (*w*/*v*) glucose concentration as a carbon source. After incubation at 45 °C for 48 h, the CO_2_ gas trapped within Durham tubes was presumptively noticed as positive [32].

#### 2.4.3. Screening of Polysaccharide Degrading Enzyme Production

To evaluate the production of polysaccharide degrading enzymes, a single colony from each yeast isolate was selected and spotted on yeast malt extract (YM) agar containing various polysaccharides as the sole carbon sources including 0.5% (*w*/*v*) carboxymethyl cellulose (CMC), locust bean gum (LBG), pectin and beech wood xylan, along with the supplementation of 0.01% (*w*/*v*) of trypan blue to indicate the presence of a clear zone [33]. Starch was also included as the tested carbon source, but no trypan blue was supplemented. The formation of halos around the yeast colonies was observed after incubating at 45 °C for 3 days. For assessment of starch hydrolysis, the cultured agar plates were flooded with Lugol’s iodine solution for 30 min. After draining off the excess iodine solution, the presence of a clear zone surrounding the yeast colony against a blue-black background indicated the action capability of starch hydrolysis. The observations were based on previous studies described by Pranay et al. [34].

#### 2.4.4. Colony and Morphological Characteristics

Yeast colonies were isolated according to their morphological characteristics and purified by streaking on yeast malt extract (YM) agar. Yeasts morphologies were observed under microscope [34].

#### 2.4.5. Molecular Identification

The selected yeast was cultured overnight in yeast malt extract (YM) broth at 37 °C with shaking at 150 rpm. Genomic DNA was extracted following the guidelines provided by the manufacturer (Promega Corp., Madison, WI, USA). Polymerase chain reaction (PCR) was performed to amplify the D1/D2 domain of the large subunit (LSU) rRNA gene by two universal primers pair NL1 (5′-GCATATCAATAAGCGGAGGAAAAG-3′) and NL4 (5′-GGTCCGTGTTTCAAGACGG-3′), following the method described by Kurtzman and Robnett [35]. The amplified DNA was then purified and sent for sequencing to a service provider (1st BASE Pte Ltd., Singapore). The gene sequence was analyzed using the BLAST algorithm of GenBank (http://www.ncbi.nlm.nih.gov/blast, accessed on 16 January 2022).

### 2.5. Investigation of β-Glucosidase and CMCase Production by Cyberlindnera rhodanensis DK

A seed culture of the selected strain *C. rhodanensis* DK 2% (*v*/*v*) was transferred into 50 mL of yeast malt extract (YM) broth containing 0.5% (*w*/*v*) CMC. The mixture was then incubated at two different temperatures, 37 °C and 45 °C, for 24 h. At three time points 0, 12 and 24 h, the culture was sampled for analysis during the incubation period. The withdrawn samples were centrifuged at 12,000 rpm, 4 °C, and 10 min to separate supernatants from cell pellets. The supernatants were collected and assayed for activities of extracellular β-glucosidase and CMCase enzymes. The cell pellets were washed twice with 1 mL of 100 mM citrate phosphate buffer and suspended in 1 mL of the same buffer. The yeast cell suspension was disrupted using an ultrasound crusher (Sonics and Materials Inc., Newtown, CT, USA), a VC 63 (Vibra Cell^TM^) operating at 100 W, 22.5 kHz. The disintegration process involved a working time of 5 s followed by a stop time of 9 s, with a total working time of 12 min. The disrupted yeast cells suspension was centrifuged at 12,000 rpm and 4 °C for 10 min. The resulting supernatants were assayed for the activities of intracellular β-glucosidase and CMCase, as described by Wang et al. [24].

The *p*NPG and 0.5% (*w*/*v*) of CMC were used as substrates for β-glucosidase and CMCase, respectively. Briefly, 50 µL of enzyme supernatant was mixed with 50 µL of 4 mM *p*NPG to assay for β-glucosidase activity. The mixture was independently incubated at 37 °C and 45 °C, respectively, for 30 min. The enzyme reaction was halted by introducing 100 µL 50 mM Na_2_CO_3_. Subsequently, the absorbance of the final reaction mixture was measured at 405 nm by a spectrophotometer (Metertech SP-8001 UV/Visible Spectrophotometer, Metertech Inc., Taipei, Taiwan). A blank solution containing 100 mM of citrate phosphate buffer (pH 6.0) was used as a reference for the substrate and enzyme blanks as well [36]. For the CMCase assay, 125 µL of supernatant was mixed with 125 µL of 0.5% (*w*/*v*) CMC. The mixture was incubated at 37 °C and 45 °C, respectively, for 20 min. Then, 250 µL DNS was added followed by boiling for 10 min. Two mL of dH_2_O was added after cooling, and the absorbance was measured at 405 nm. The unit of enzyme was defined as the amount of enzyme liberating 1 micromole of product per minute per mL under the assayed conditions [37].

### 2.6. Screening of Carbon Sources for Extracellular β-Glucosidase Production by C. rhodanensis DK

A variety of carbon sources (glucose, xylose, fructose, cellobiose, maltose sucrose, lactose, xylan, CMC, LBG and defatted rice bran) were investigated to assess their potential as alternative inducers or substrates for extracellular β-glucosidase production. A seed culture of the selected yeast strain at 2% (*v*/*v*) was introduced into yeast malt extract (YM) broth supplemented with different carbon sources. The cultures were incubated at 37 °C under continuous agitation at 150 rpm for 48 h. Samples were collected at 0, 24 and 48 h and subsequently centrifuged at 12,000 rpm and 4 °C for 10 min. The supernatants obtained after centrifugation were examined for extracellular β-glucosidase activity using the method described earlier [36].

### 2.7. Confirmation of the Main Glycosiase Activities in Culture Supernatant of C. rhodanensis DK

In this study, a seed culture, comprising 2% (*v*/*v*) of the selected strain DK, was introduced into yeast malt extract (YM) broth supplemented with xylose as a sole carbon source. The culture was incubated at 37 °C under continuous agitation at 150 rpm for 48 h. The culture broth was centrifuged by 10,000 rpm at 4 °C for 10 min, and the cell free culture supernatant was used to determine the enzyme activity of β-glucosidase, β-xylosidase and β-mannosidase using 4.0 mM of *p*NPG (para-nitrophenol-β-D-glucopyranoside (PNP-β-D-glucopyranoside: *p*NPG), 2.0 mM of *p*NPX (PNP-β-D-xylopyranoside: *p*NPX) and 2.0 mM of *p*NPM (PNP-β-D-mannopyranoside: *p*NPM) as the specific substrates, respectively. The supernatants obtained after centrifugation were assayed for extracellular enzyme activity as mentioned above [38].

### 2.8. Comparison of Extracellular β-Glucosidase Production of C. rhodanensis DK with the Reference Strains

A seed culture 2% (*v*/*v*) of *C. rhodanensis* reference strains (TBRC-BCC 64435 and A45.3) and DK were individually inoculated into yeast malt extract (YM) broth supplemented with either 1% (*w*/*v*) xylose or xylan. The cultures were incubated at 37 °C and 45 °C under continuous agitation at 150 rpm for 48 h. Samples were collected at 0, 24 and 48 h, and the cultures were centrifuged at 12,000 rpm for 10 min at 4 °C. The supernatants were then used to examine the extracellular β-glucosidase activity at 45 °C, following the method described by Wang et al. [24].

### 2.9. Statistical Medium Optimization for Extracellular β-Glucosidase Production Using Xylose and/or Xylan as the Sole Carbon Source

#### 2.9.1. Plackett–Burman Design (PBD)

The important medium components influencing extracellular β-glucosidase production were screened using PBD. In this trial, a 15-run PBD was used to evaluate six factors including yeast extract, peptone, malt extract, MgSO_4_, KH_2_PO_4_, xylose or xylan. Each independent variable was examined at two levels: +1 for the high level and −1 for the low level, and center points were also run to evaluate the linear and curvature effects of the variables. The fermentation medium was formulated according to the experimental design. The pH of the cultures was adjusted to 6.0, and the cultures were incubated at 37 °C 150 rpm for 48 h. This analysis was carried out using the statistical software package Design-Expert 7.0.

#### 2.9.2. Central Composite Design (CCD) and Response Surface Methodology Analysis (RSM)

To determine the optimum level of nutritional components for the highest enzyme production, the significant influencing factors indicated by PBD results were selected to optimize using the central composite design (CCD) incorporated with response surface methodology. A statistical design matrix was established, comprising three levels for each variable: A represented yeast extract, B represented peptone and C represented either xylose or xylan. Different concentrations of the nutritional factors obtained from PBD screening were prepared according to the actual values of the factor levels specified by the coded values. The levels included in the matrix were (−1, +1), axial point (−α, +α) and center point (0). The obtained experimental data were subjected to analysis using a second-order polynomial equation. The data were fitted to the quadratic model through the utilization of Design-Expert 7.0 software. This approach allowed the researchers to develop a mathematical equation that described the relationship between the variables and the response (β-glucosidase production), enabling them to draw meaningful conclusions and optimize the production process. A second-order polynomial equation is described as follows:$$Y=\beta_{0}+\sum_{i=1}^{2}\beta_{i}X_{1i}+\sum_{i=1}^{2}\beta_{ii}X_{i}^{2}+\sum_{i=1}^{1}\sum_{j=i+1}^{1}\;\beta_{ij}X_{i}X_{j}\;$$
where Y is the dependent response, *Xi* and *Xj* are the independent factors and $\;\beta_{0}$, $\beta_{i}$ and $\beta_{ij}$ are the model coefficients that were obtained using the linear least squares method. The optimization experiment was validated by assays for extracellular β-glucosidase activity as described in (2.5) at 37 °C, pH 6.0 and samples were systematically collected at regular 6 h intervals over a period of 72 h [24].

### 2.10. Thermostability Test

To assess thermostability, a crude enzyme solution was prepared using a modified method, as described by Yadav et al. [39]. In this experiment, the crude enzyme was divided into two portions: one portion received the addition of a protease inhibitor cocktail sourced from Sigma-Aldrich (St. Louis, MO, USA), while the other portion remained without any protease inhibitor. The total volume of the reaction (2 mL) including 1.8 mL of the crude enzyme was mixed with 0.2 mL of protease inhibitors or citrate-phosphate buffer (100 mM). Samples were incubated at 35 °C, 45 °C and 55 °C for 3 h. Thereafter, the remaining enzyme activity was measured and presented as the relative residual activity (%).

### 2.11. Statistical Analysis

The statistical values, including the mean ± standard deviation (SD), were derived from three separate and independent trials. These values were obtained using a one-way analysis of variance (ANOVA) followed by a post hoc test (Tukey) using the IBM Statistics SPSS software 20. The significance level for all analyses was set at *p* ≤ 0.05.

## 3. Results and Discussion

### 3.1. Isolation of Thermotolerant Yeasts

In this study, a collection of 18 yeast isolates were obtained from eight different samples of Laphet-so. Twelve yeast isolates were obtained from the sample collected from Myaadia village, five isolates were achieved from a Tanngyo local Laphet-so producer, one isolate showed growth from Taunggyi local market and no yeast isolate was achieved from other samples. The focus of our investigation was on identifying thermotolerant yeasts from various samples across different regions, crucial for their potential in industrial applications. Notably, yeasts capable of thriving at 40 °C were considered thermotolerant in our study [40]. To situate our findings, several studies have identified the significance of natural fermented foods, soil and plants as promising sources for isolating thermotolerant yeasts. Talukder et al. [41] successfully isolated twenty-five thermotolerant yeast strains from natural fermented sources in Bangladesh. Most of these yeast demonstrated the ability to grow within the temperature range of 45–50 °C. Additionally, Choi et al. [42] reported the isolation of thermotoletant yeast from Nuruk, a traditional Korean fermentation starter, with *Pichia kudriavzwvii* demonstrating rapid growth at 44 °C. Whereas the isolation of 18 and 14 thermotolerant yeasts capable of growth at least 40 and 45 °C from various natural fermented sources collected from various locations in Morocco was also reported [40]. Furthermore, Kaewkrajay et al. [43] reported 2 yeast strains that were capable of growth at 50 °C; 169 strains grew at 45 °C, and 96 strains were able to grow at 40 °C. Our findings align with these studies, collectively emphasizing the potential of diverse environmental sources, including traditional fermented foods like Laphet-so, in yielding thermotolerant yeasts with industrial significance. The variation in thermotolerant yeast populations across distinct Laphet-so samples enriches our comprehension of regional differences.

### 3.2. Characterization of Thermotolerant Yeasts

#### 3.2.1. Evaluation of Tannin-Tolerant Ability

A total of 18 yeast isolates were able to grow ranging from 0.5 to 3% (*w*/*v*) of tannic acid concentrations incubation at 45 °C for 3 days. However, at a higher tannic acid concentration of 5% (*w*/*v*), only six isolates (TN5, MD3, MD6, MD12, MD14 and MD20) were able to grow. The result is shown in Table 1. Tannins are water-soluble polyphenolic compounds that belong to a category of recalcitrant molecules that are resistant to microbial attacks and toxic to various microorganisms [44]. Microorganisms capable of growing in tannins-rich environments possess specialized metabolic mechanisms to tolerate the toxicity of tannins. In this context, the ability of yeast strains isolated from Laphet-so were able to resist content at a tannic acid concentration of 50 g/L. Our findings align with the results of studies conducted by Kanpiengjai et al. [7], Kanpiengjai et al. [45], Pan et al. [46], Song et al. [47] and Zhang et al. [48], where yeast strains exhibited varying degrees of tannin tolerance from different environmental sources. The observed tannin tolerance of these yeast strains aligns with previous research by Kanpiengjai et al. [7], who studied yeast isolates associated with Miang (fermented tea leaves) from north Thailand. Their findings suggested that yeasts originating from tannin-rich environments may possess additional response mechanisms to overcome the adverse or toxic effects of tannins. Additionally, Kanpiengjai et al. [45] reported on yeasts from *Camellia sinensis* var. *assamica* flower, with the ability to tolerate a concentration of 50 g/L tannic acid. Furthermore, Pan et al. [46] reported that the yeast *Rhodosporidium diobovatum* isolated from mangroves is capable of growing in the presence of tannic acid concentrations as high as 25 g/L. Moreover, Song et al. [47] isolated 13 tannin-tolerant yeast strains from mangrove environments that thrived at a tannic acid concentration of 15 g/L, while Zhang et al. [48] obtained the yeast *Aureobasidium melanogenum* from a red wine starter culture, which demonstrated the ability to tolerate a tannic acid concentration of 20 g/L. The observed strong tannin tolerance exhibited by our yeast isolates has significant industrial potential, particularly in the sectors involved with high tannin content substrates. This resilience may prove crucial in applications dealing with tannin-rich waste streams or bioremediation efforts.

#### 3.2.2. Ethanol Producing Capability

The results of our study indicate that among the 18 yeast isolates, only the DK strain exhibited the capacity to produce ethanol at elevate temperatures. Notably, none of the other strains yielded ethanol at 45 °C for a two-day incubation period, as evidenced by the absence of gas formation (Table 1). This result highlights the unique thermotolerance and ethanol-producing potentiality of the DK strain, emphasizing its significance in the domains of biofuel and industrial fermentation applications [49]. In the context of biofuel and ethanol production from cellulosic materials, it is crucial to identify microorganisms that can perform simultaneous saccharification and fermentation at high temperatures [14]. Notably, a study in Bangladesh involving 18 thermotolerant yeast strains from fermented sources highlighted that 8 displayed the ability to produce bioethanol [41]. Furthermore, Techaparin et al. [50] conducted a comprehensive study on various thermotolerant yeasts, including strains of *Saccharomyces cerevisiae* and *Pichia kudriavzevii*. Their research revealed noteworthy capabilities in terms of both high-temperature tolerance and ethanol production, particularly at a temperature of 45 °C. Additionally, Phong et al. [32] isolated the thermotolerant yeast *Pichia kudriavzevii* from soil samples collected from the Mekong Delta, Vietnam, which exhibited significant potential for high-temperature ethanol fermentation at 45 °C. These collective findings contribute to our understanding of the diverse and promising avenues for thermotolerant yeast utilization in high-temperature ethanol production and industrial applications.

#### 3.2.3. Production of Polysaccharide Degrading Enzymes

In this study, to assess the functional capabilities of yeast in producing extracellular polysaccharide degrading enzymes such as cellulases, β-mannanase, pectinase, xylanase and amylase, an agar plate screening method was used to observe the clear zone formation. The findings from our study indicate that all yeast isolates demonstrated the capability to degrade yeast malt extract (YM) agar when provided with the substrate CMC, resulting in the formation of distinct clear halo zones. This outcome suggests that these yeast strains possess the enzymatic machinery necessary to hydrolyze cellulose, specifically CMC. However, when yeast malt extract (YM) agar was supplemented with other substrates such as LBG, pectin, xylan and starch, no clear zone formation was observed after a 3-day incubation period at 45 °C. This absence of a clear zone suggests that these thermotolerant yeasts isolates did not exhibit significant enzymatic activity against these particular substrates under the tested conditions. The formation of clear zones observed during the degradation of CMC highlights the remarkable ability of thermotolerant yeasts to produce extracellular cellulases enzymes, as demonstrated in Table 1. The appearance of clear zones on CMC agar indicates that these yeasts possess pathways for converting complex polysaccharide, as reported by Waghmare et al. [51]. This finding is consistent with the results obtained by Delgado-Ospina et al. [52], who explored the functional biodiversity of yeasts isolated from Colombian fermented and dry cocoa beans, highlighting their thermotolerance and cellulases production capabilities. Similarly, Rai et al. [53] reported that thermotolerant yeast *Candida* spp. exhibited maximum clear zones on the cellulose-containing media, accompanied by significant cellulases enzyme production. Furthermore, Adelabu et al. [54] successfully isolated yeasts that tested positive for cellulase production, as evidenced by clear zones on CMC agar. The cellulases constitute a vital group of enzymes crucial for the breakdown of cellulose, a complex polysaccharide present in plant cell walls. This information underscores the potential of thermotolerance and their cellulases-producing abilities by these yeasts, which could unlock innovative solutions for various industrial processes. Additionally, investigating the potential synergies between different enzymes produced by these thermotolerant yeasts could pave the way for more efficient and versatile enzymatic applications.

#### 3.2.4. Colony and Morphological Characteristics

The colony and morphological characteristics of the DK yeast strain were observed, and we found that the growth on yeast malt extract (YM) agar media possessed a white-creamy, convex, round shape with filamentous margins, consisting of circular or ovoid budding yeast cells (Figure 1). The recognition of these typical colony morphologies is essential for clinical diagnosis. Scientific and clinical laboratories often use the colony morphology as an adjunct to identify microbial species based on their distinct growth patterns, a practice corroborated by Sousa et al. [55]. In the context of our study, we meticulously documented the colony and morphological characteristics of the DK yeast strain while incubated at 45 °C for 2 days. This comprehensive examination of yeast traits contributes significantly to our understanding of their behavior and potential applications across various fields.

#### 3.2.5. Molecular Identification

In our efforts to identify the DK strain, we employed sequence analysis targeting the D1/D2 domain of the LSU rRNA gene. The resulting sequence was then subjected to a BLAST search, which unveiled a remarkable similarity of 99.83% with the type strain of *Cyberlindnera rhodanensis*. This high degree of sequence similarity observed identifies the yeast isolate DK as a member of the *C. rhodanensis* species. The LSU rRNA gene, specifically its D1/D2 domain, is a widely recognized molecular marker for the precise identification and classification of yeasts and fungi [56]. *C. rhodanensis* is known for its diverse ecological and industrial significance, particularly in the context of ethanol fermentation under thermotolerant conditions, as highlighted by the finding of Keo-oudone et al. [29]. Our research aligns with theirs, further supporting the yeast’s capacity to thrive and produce ethanol at elevated temperatures. This characteristic is of great interest due to its potential applications in industrial ethanol production. Additionally, it is noteworthy that *C. rhodanensis* (specifically A22.3 and A45.3) has been found to produce tannase, as previously isolated from the Miang sample [30]. This discovery broadens the potential industrial applications of *C. rhodanensis* as tannase has various uses in food processing and pharmaceuticals. In addition, the study by Kodchasee et al. [10] revealed that *C. rhodanensis* displays the dual capability of not only synthesizing the β-glucosidase enzyme but also producing esters, which are essential components in the process of wine production. The metagenomics analysis reported by Bo et al. [4] revealed that the yeast *Cyberlindnera* sp. was identified as a predominant genus in Laphet, further underlining the ecological diversity of this yeast group and its potential role in various environments. This ecological insight prompts further exploration into the adaptive strategies and functional roles of *Cyberlindnera* sp. in various ecological niches. This yeast *C. rhodanensis*, known for its ethanol production, enzymatic capabilities and ecological prevalence, opens avenues for targeting applications in biotechnology and environmental studies.

### 3.3. Investigation of β-Glucosidase and CMCase Production by C. rhodanensis DK

In this study, we investigated the activities of β-glucosidase and CMCase in both the cell-free supernatant of a yeast cell culture and intact yeast cells during fermentation at temperatures of 37 and 45 °C. Enzyme activities were measured at regular intervals of 0, 12 and 24 h (Figure 2). At 37 °C of enzyme production and reaction, intracellular β-glucosidase and CMCase activities were observed with the low values of 0.06 and 0.01 U, respectively. In contrast, at 45 °C, intracellular β-glucosidase and CMCase activities found up to 0.08 and 0.003 U, respectively. Conversely, when yeast cells were cultivated at 45 °C, intracellular β-glucosidase and CMCase activities were found with the values of 0.003 and 0.001 U, respectively, when the enzyme reaction was carried out at 45 °C. Extracellular β-glucosidase and CMCase activities were 0.97 and 0.49 U, respectively, when the enzyme reaction was carried out at 37 °C. At 45 °C enzyme reaction, extracellular β-glucosidase and CMCase activities were 1.70 and 0.90 U, respectively. When yeast cells were cultivated at 45 °C, the enzymes activities were achieved with values of 0.98 and 0.22 U, respectively, at 37 °C. However, after 24 h of incubation at 45 °C, the enzymes activities were higher at 1.41 and 0.63 U, respectively.

The investigation into the intracellular and extracellular activities of the enzymes CMCase (cellulase) and β-glucosidase at different temperatures is of importance in understanding the metabolic processes of microorganisms. Through the study of these enzymes, valuable insights can be achieved into the functioning of microorganism metabolic pathways, thereby enhancing our comprehension of its overall physiology, as highlighted by Burns and Dick [57]. Furthermore, the identification and utilization of these enzymes hold the potential to drive the development of more efficient and sustainable processes. Cellulose, a linear polymer of glucose, serves as the substrate for cellulases, including CMCase and β-glucosidase. The differentiation between CMCase and β-glucosidase is crucial due to their distinct roles. CMCase, encomposing endoglucanases and exoglucanases, catalyze the random hydrolysis of cellulose, generating mainly cellobioses. In contrast, β-glucosidase acts to further hydrolyze cellobiose into free glucose molecules, as indicated by Saritha Mohanram [58]. The experimental results demonstrate that *C. rhodanensis* DK exhibited extracellular activity of both CMCase and β-glucosidase enzymes; among them, the highest was extracellular β-glucosidase activity. This finding signifies the remarkable ability of *C. rhodanensis* DK to produce extracellular β-glucosidase. Similarly, Chuengcharoenphanich et al. [59] also noted that *C. rhodanensis* possesses the capability to produce endoglucanase, exoglucanase and β-glucosidase enzymes. Furthermore, the work of Kodchasee et al. [10] supports this, highlighting that *C. rhodanensis* isolated from Miang has the capacity to produce extracellular β-glucosidase enzymes. Additionally, it is noteworthy that at both 37 and 45 °C during enzyme production, yeast cells presented higher extracellular β-glucosidase and CMCase activities. Among the enzyme activities observed, β-glucosidase showed the highest activity in yeast cell culture at 37 °C, while the enzyme also reacted with the substrate at 45 °C. Temperature significantly influences microbial growth and enzyme production. Yeasts thrive within specific temperature ranges, and deviations can impact their metabolism [30,60]. In our study, β-glucosidase presented higher activity at 37 °C in yeast cells, likely due to optimal growth conditions promoting enzyme synthesis extracellular functions. Additionally, the observed increase in enzyme activity could be linked to changes in the structure and stability of enzymes at higher temperatures. Some enzymes are known to exhibit higher activity at elevated temperatures due to enhanced substrate binding and catalytic efficiency [61]. Numerous studies have delved into the specific mechanisms that underlie the temperature-dependent behavior of enzymes. Somero [62] demonstrates how changes in temperature influence enzyme-substrate interactions and catalytic efficiency with the Arrhenius equation, which elucidates the relationship between temperature and reaction rates. The results of this study provide valuable insights into the temperature-dependent dynamics of β-glucosidase and CMCase activities, emphasizing the dynamic nature of *C. rhodanensis* DK metabolic responses during fermentation.

### 3.4. Screening of Different Carbon Sources for Extracellular β-Glucosidase Production by C. rhodanensis DK

*C. rhodanensis* DK was assessed for its extracellular β-glucosidase production, utilizing a wide range of carbon source sugars, including monomers, dimers, polymers and defatted rice bran. Enzyme activity was observed to vary based on the carbon sources, as presented in Figure 3. The highest β-glucosidase activity was achieved when *C. rhodanensis* DK was cultivated on xylose and xylan, resulting in activity of 3.07 ± 0.04 U/mL and 3.08 ± 0.1 U/mL, respectively. This was followed with cellobiose (1.11 ± 0.01 U/mL) and maltose (0.38 ± 0.02 U/mL) as inducers. The findings from our study have revealed notable variations in β-glucosidase activity, which are contingent on the carbon sources used for cultivation. The highest activity was recorded when xylose and xylan were employed as inducers. Interestingly, this observation aligns with the research of Sørensen et al. [36], who reported that β-glucosidase enzyme activity of *A. saccharolytic* growth on a solid medium displayed an increase when cultured with xylan (0.7 U/mL) and xylose (0.8 U/mL) as the sole carbon sources. Furthermore, it is notable that β-glucosidase, which exhibits tolerance to and stimulation by xylose, has been extensively documented in previous studies [36,63]. The observed variations in β-glucosidase activity highlight the complexity of enzyme induction processes. Enzyme induction represents a complex process involving various molecular changes that ultimately culminate in the production of specific enzymes necessary for the degradation and utilization of different carbon sources [36]. In this regulatory mechanism, a substrate (or a structurally related compound or metabolically linked compound) triggers the synthesis of enzymes that are typically involved in breaking down that specific substrate. These enzymes, synthesized when specific genes are activated, are referred to as inducible enzymes, and the chemical responsible for initiating gene transcription is called the inducer. Inducible enzymes are only produced in response to the presence of their respective substrate, ensuring that they are generated as needed [64]. Additionally, similar to enzyme induction, carbon source regulation, commonly referred to as carbon catabolite repression (CCR), serves as a conservative mechanism that safeguards a cell protein synthesis machinery. Carbon catabolite repression is primarily triggered by the presence of glucose, but it is important to note that in various organisms, other rapidly metabolized carbon sources can also induce this repression. In some instances, these alternative carbon sources can even inhibit the breakdown of glucose [64]. An illustrative example of this phenomenon can be found in the work of Zanoelo et al. [65], who explored the effects of varying concentrations of glucose and xylose on enzyme activity. They noted that both xylose and glucose may act at the same binding site of the *Scytalidium thermophilum* β-glucosidase, while Corrêa et al. [63] demonstrated a mechanism by which xylose stimulates β-glucosidase activity. They examined a glycosidase hydrolase 1 family β-glucosidase from *Thermotoga petrophila* (TpBg11), which demonstrated a typical monosaccharide stimulation mechanism. Specifically, when glucose or xylose binds to the binding regions (BR1), an adjacent loop region adopts an extended conformation, thereby facilitating increased access to the TpBg11 active site. This structural change enhances product formation, shedding light on the intricate regulatory mechanisms underlying β-glucosidase stimulation by xylose. The finding contributes valuable insights into the regulatory mechanisms orchestrating β-glucosidase activity in response to specific carbon sources.

### 3.5. Confirmation of the Main β-Glucosidase Activities Produced by C. rhodanensis DK

Our investigation into the specific enzyme activities of the main β-glucosidase produced by *C. rhodanensis* DK, with xylose or xylan as inducer, sheds light on the substrate specificity and catalytic preferences of this yeast strain. The results of this study showed that when the substrate *p*NPG was used, the enzyme activity was detected in the same level as found in the enzyme production with our earlier observations during the carbon source screening experiment. This alignment underscores the efficiency of *C. rhodanensis* DK in hydrolyzing β-glucosidic bonds within the *p*NPG molecule. Interestingly, our results indicated no enzymatic activity was detected when *p*NPX was employed as a substrate. This finding aligns with earlier research reported by Bonfá et al. [66], which established that *C. rhodanensis* DK indeed produces β-glucosidase with the capability of efficient hydrolyzing β-glucosidic bonds without cross-activity on the β-xylosidic bond. The specificity of the synthesized β-glucosidase further emphasizes the selectivity of the enzymatic machinery employed by *C. rhodanensis* DK. Our finding aligns with research conducted by Bhalla et al. [67] on thermostable β-xylosidase from *Geobacillus* sp. that suggests that some particular microbial strains may lack the enzymatic machinery necessary for β-xylosidase production or that the experimental conditions tested did not favor its expression. Similarly, when exploration of a different substrate, *p*NPM, was employed, no enzymatic activity was observed. This result indicates that *C. rhodanensis* DK does not produce detectable levels of β-mannosidase activity under the conditions tested. Remarkably, in this study, it was discovered that β-glucosidase from *C. rhodanensis* DK is induced by xylose and xylan, as evident by the screening of carbon sources. Consequently, this study not only confirms the substrate specificity of the synthesized β-glucosidase but also highlights the specificity of enzymatic induction and specificity in *C. rhodanensis* DK.

### 3.6. Comparison of Extracellular β-Glucosidase Production by C. rhodanensis DK with the Reference Strains

Three strains of *C. rhodanensis* were further studied on extracellular β-glucosidase productions at 37 and 45 °C in culture medium containing 1% (*w*/*v*) of either xylose or xylan, and the results are presented in Figure 4. As reported by the results of enzyme production at 37 °C and 48 h fermentation, the DK strain produced 3.08 ± 0.01 and 3.11 ± 0.1 U/mL of the β-glucosidase activities with xylose and xylan, respectively. The enzyme produced by the strain TBRC-BCC 64435 increased to 2.15 ± 0.01 U/mL accompanied by xylan, while using xylose induction produced only 0.22 ± 0.01 U/mL. In the case of the strain A45.3, xylose and xylan inductions showed only 0.23 ± 0.01 and 0.22 ± 0.01 U/mL, respectively. Although the β-glucosidase productions of *C. rhodanensis* A45.3 and *C. rhodanensis* TBRC-BCC 64435 were lower than the DK strain, the trend of cell growth was almost the same. This result confirms the difference in enzyme synthesis in cellular response to external inducers between the newly isolated *C. rhodanensis* DK and the reference strains. In the culture conditions for enzyme production at 45 °C for 48 h, the *C. rhodanensis* DK strain showed the highest β-glucosidase among *C. rhodanensis* strains as the enzyme activities of 0.55 ± 0.01 and 0.75 ± 0.01 U/mL were detected when xylose and xylan were used as inducers, respectively, while *C. rhodanensis* TBRC-BCC 64435 produced 0.03 ± 0.01 and 0.70 ± 0.01 U/mL by either xylose or xylan, respectively. The strain A45.3 produced the enzyme 0.02 ± 0.01 and 0.02 ± 0.01 U/mL when xylose and xylan were used as the substrates.

The results of our study reveal important insights into the extracellular β-glucosidase production of *C. rhodanensis* DK, in comparison to the reference strains. We explored the impact of both temperature and carbon source on β-glucosidase activity, shedding light on the unique characteristics of these strains in response to different growth conditions. At 37 °C, the DK strain exhibited significantly higher β-glucosidase activity when induced with xylose and xylan, outperforming the reference strains TBRC_BCC 64435 and A45.3. However, when we shifted our focus to the 45 °C culture condition, *C. rhodanensis* DK continued to lead in β-glucosidase production. This excellent performance demonstrates that *C. rhodanensis* DK is not only highly responsive to inducers but also remarkably adaptable to elevated temperatures. Our findings align with previous research, as observed by Masoud and Jespersen [68], who reported variations in enzyme secretion within yeast species, highlighting that these variations are not unique to *C. rhodanensis* but rather represent a common feature of microbial diversity. Furthermore, our observations regarding the impact of temperature on enzyme production shows that temperature serves as a significant determinant of microbial metabolism [69]. The variation in enzyme production observed in our study can be attributed to several factors, including induction and regulation, genetic variations in different environmental and cultivation methods. These factors influence gene expression and enzyme production in microorganisms, resulting in variations in enzyme activities within the same species [70,71].

### 3.7. Statistical Medium Optimization for Extracellular β-Glucosidase Production Using Xylose and/or Xylan as the Carbon Sources

#### 3.7.1. Plackett–Burman Design (PBD)

To determine the optimal parameters for enhancing enzyme production by the yeast strain *C. rhodanensis* DK, using a modified YM broth medium, we employed the Plackett–Burman design. This experimental approach allowed us to investigate the effects of six factors within the medium ingredients on β-glucosidase enzyme production. The results are presented in Table 2, utilizing xylose and xylan as inducers. Upon analyzing the design matrix, we identified the key factors that significantly influenced β-glucosidase enzyme production. Specifically, yeast extract, peptone and the presence of xylose and/or xylan were found to be significant (*p* < 0.05) after 48 h of fermentation. The R^2^ values for xylose and xylan were calculated to be 0.8016 and 0.9188, respectively. These values indicate that up to 80% of the variability in enzyme production can be accounted for by the model, as shown in Table 3. It is remarkable that these screening studies led to a substantial increase in enzyme production. The Plackett–Burman design is a valuable statistical design that is particularly useful when dealing with a large number of potential factors that may influence the targeted product yield. Based on the employment of linear regression analysis, this design helps to identify and prioritize the significant factors affecting the responses of interested products while filtering out irrelevant factors [72]. However, the PBD presents certain limitations. Strong confounding arises between its main factors and any potential two-factor interactions, making it challenging to discern between main effects and interactions. Moreover, this design approach lacks the capability to estimate interactions for factors with more than two levels [73].

#### 3.7.2. Central Composite Design (CCD) and Response Surface Methodology Analysis (RSM)

The three significant variables identified from the Plackett–Burman design results were further optimized using CCD and RSM. CCD was employed to determine the optimal concentrations of these significant variables for achieving maximum β-glucosidase production. The CCD matrix, which is characterized in actual coded units in Table 4 and Table 5, includes the actual observed results of the responses. This experimental design allowed for a systematic exploration of the effects of these factors and their combined influence on the response, facilitating the identification of optimal conditions and concentrations for maximizing β-glucosidase production. RSM played an essential role in exploring the interactions among the significant factors. The substantial effects of the interactions between the three variables were illustrated through the creation of 3D contour plots, presented in Figure 5. These plots were generated by examining all possible combinations of the independent variables in the study.

The quadratic equations were used to elucidate the combined influence of these variables and their concentrations’ influence on the response. For instance, the equation for xylose and xylan substrates, obtained from coefficient effects, is as follows:β-Glucosidase activity (U/mL) = 7.09 + 0.21A + 0.53B + 0.10C − 0.30AB + 0.45AC + 0.68BC − 0.71A^2^ − 0.79B^2^ − 1.65C^2^
where, A = yeast extract, B = peptone and C = xylose

Notably, the positive coefficients of A and C indicate that β-glucosidase activity increases with higher levels of yeast extract and xylose. Similarly, for xylan as an inducer, the equation is
β-Glucosidase activity (U/mL) = 9.86 + 0.088A + 1.26B + 1.93C + 0.12AB − 0.13AC + 0.99BC − 1.38A^2^ − 0.54B^2^ − 1.67C^2^
where A = yeast extract, B = peptone and C = xylan, respectively.

In this case, the positive coefficients of B and C suggest that β-glucosidase activity increases with higher levels of peptone and xylan.

Furthermore, the combined influence of these variables on the response was statistically evaluated using ANOVA. This analysis yielded a *p*-value of 0.0072 with an R^2^ value of 0.8299 for the xylose inducer and a *p*-value of 0.0001 with an R^2^ value of 0.9759 for the xylan inducer. These results indicate that the model is statistically significant and effectively explains the observed variations in β-glucosidase activity under the influence of these variables. Through solving equations and analyzing RSM plots, the optimized medium levels for β-glucosidase production was determined to be yeast extract at 5.83 g/L, peptone at 10.81 g/L and xylose at 20.20 g/L for xylose as an inducer. Similarly, when xylan was used as an inducer, the maximum β-glucosidase production was achieved with yeast extract at 5.79 g/L, peptone at 13.68 g/L and xylan at 20.16 g/L.

To validate the experimental model, the experiments were conducted using a statistically optimized medium. The practical response produced a value of 6.73 ± 0.08 U/mL, which is about 93.5% of the predicted value when xylose was used as an inducer. Similarly, the practical production of β-glucosidase was 8.92 ± 0.05 U/mL, approximately 90.4% of the predicted value when xylan was used as an inducer (Figure 6). These results demonstrate a close agreement between the experimental and predicted values, indicating the successful optimization of the medium composition using the experimental design.

In this investigation, PBD and RSM were employed to identify the critical factors affecting β-glucosidase production by *C. rhodanensis* DK. Numerous research reports have documented the application of these methods for medium optimization with various microorganims, including *Zobellella denitrificans* [74], *A. versicolor* [75] and *B. stratosphericus* [76]. Based on the findings from PBD, it was evident that among the six component factors considered, yeast extract, peptone and xylose/xylan had a statistically significant effect on enzyme production. These findings are consistent with earlier research conducted by Nanjundaswamy and Okeke [77], which also emphasized the stimulatory effect of peptone and yeast extract on β-glucosidase production. Additionally, Sørensen et al. [36] reported that β-glucosidase activity from *A. saccharolyticus* was achieved in the highest activity in the cultures grown in a medium containing xylose and xylan as carbon sources. The alignment of our findings with these previous studies further supports the importance of these variables in influencing β-glucosidase production in this current research. Furthermore, the implementation of CCD and RSM indicated the optimal concentrations of variables for β-glucosidase enzyme production. Tasharrofi et al. [78] noted that low nutrient concentrations often fail to induce the target enzyme production in most microorganisms. Conversely, excessively high nutrient concentration can lead to a reduction in target enzyme production due to thickening of the fermentation medium, which can interfere with proper agitation and aeration, ultimately reducing air supply [79]. Additionally, similar studies have utilized this statistical experimental design for optimizing culture conditions in β-glucosidase production [80]. Through the application of this experimental design, we successfully enhanced β-glucosidase production by *C. rhodanensis* DK to achieve values of 6.73 ± 0.08 U/mL and 8.92 ± 0.05 U/mL with the respective inducers. The strong agreement between the predicted response and the experimental data obtained in the laboratory underscores the suitability and effectiveness of experimental design methods for medium optimization.

### 3.8. Thermostability Test

To assess the thermostability of crude extracellular β-glucosidase produced by *C. rhodanensis* DK, we conducted a comprehensive investigation, taking into consideration the potential impact of protease activity on β-glucosidase. Our modified approach involved the addition of a protease inhibitor to elucidate its impact on enzyme stability. The results are shown in Figure 7. The enzyme exhibited remarkable stability at 35 °C with nonsignificant loss of activity after 3 h of incubation. This finding underscores the enzyme property of maintaining its functionality under moderate temperature conditions. When the crude enzyme was incubated at 45 °C, a slight decrease in enzyme activity was observed initially, but the enzyme retained a substantial portion of its initial activity, ranging from 90 to 92%, after 60 min of incubation. The enzyme continued to display excellent stability, maintaining activity levels at 87 to 88% after extended incubation for 2 and 3 h. When exposed to the challenging temperature of 55 °C, the enzyme exhibited a reduction in activity, retaining 80 to 82% of its initial activity after 60 min and 76% after 2 h, ultimately stabilizing at 75% after 3 h. This behavior aligns with previous findings, such as those of Fawzi [81], who reported that β-glucosidase from *Fusarium proliferatum* retained 60% activity at 55 °C for 1 h. Gong et al. [82] also agreed that β-glucosidase enzyme activity is quite stable at temperatures below 60 °C. The initial decline in enzyme activity at elevated temperatures may be attributed to denaturation or structural changes, resulting from the disruption of weak bonds that maintain the enzyme’s 3D structure. However, the subsequent recovery of enzyme activity may be linked to some reversible structural adjustments that occur under less extreme conditions [83]. In addition, our findings revealed that the presence or absence of protease inhibitors generated similar trends in thermostability. This suggests that protease activity includes enzymes that did not have a significant impact on β-glucosidase during the investigation of thermostability. Proteases are enzymes responsible for catalyzing the hydrolysis of peptide bonds in proteins, resulting in their degradation or the alteration of their primary structure. In most cases, proteases play a vital role in modifying protein structure [84]. However, the acceptable thermostable characteristic of β-glucosidase from *C. rhodanensis* DK must be confirmed after the enzyme was further purified.

## 4. Conclusions

In the present study, an attempt was made to isolate thermotolerant yeast from Laphet-so and observe for the potential abilities. A total of 18 thermotolerant yeast isolates from Laphet-so samples collected from different plantations in southern Shan state, Myanmar, were isolated and evaluated for their tannin tolerance, polysaccharide-degrading enzyme production and capability in ethanol fermentation at 45 °C. Among the yeast strains, *C. rhodanensis* DK exhibits high tannin-tolerant properties, is capable of ethanol fermentation, and acts as CMC degrading enzyme producer in both intracellular and extracellular compartments at 37 and 45 °C. The highest activity was found in the extracellular β-glucosidase when cultured at 37 °C. Moreover, xylose and xylan were found to be the most suitable carbon sources for the production of extracellular β-glucosidase by *C. rhodanensis* DK. Through the application of this experimental design, the Plackett–Burman design (PBD) and the central composited design (CCD), incorporated with surface response methodology (RSM), the successful enhancement of β-glucosidase production by *C. rhodanensis* DK in the optimized medium was obtained with values of 6.73 ± 0.08 U/mL and 8.92 ± 0.05 U/mL when xylose and xylan were independently used as the sole carbon sources, respectively. Furthermore, this β-glucosidase retained 100% for 3 h at 35 °C, 88% for 3 h at 45 °C and 75% stable for 3 h at 55 °C. According to this study, Laphet-so can be regarded as a source of thermotolerant yeast exhibiting industrial potential indicated by high tannin tolerance, ethanol fermentation from cellulosic modeling substrate CMC and robust enzyme production. These findings indicate that *C. rhodanensis* DK is a potential candidate yeast strain for industrial applications in the production of polysaccharide-degrading enzymes (specifically β-glucosidase), using xylose as a low-cost inducer. Additionally, this selected yeast is possibly beneficial for other food and beverage fermentations, especially those involving high tannin containing substrates. The exploration of its capabilities and the successful optimization of β-glucosidase production provide a foundation for future investigations into molecular mechanisms, further refinement of optimization strategies and exploration of enzyme purification and characterization. The observed stability of β-glucosidase at different temperatures shows the potential applications; nonetheless, further research focused on understanding its molecular basis for enhancing the enzyme thermostability and the broader industrial utilization is essentially required.

## Figures and Tables

**Figure 1 jof-10-00243-f001:**
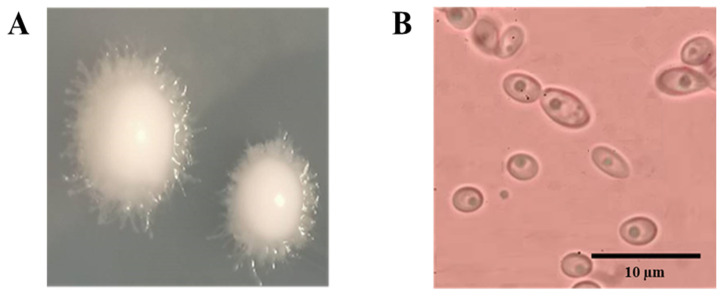
Colony morphology (**A**) and morphological characters of yeast DK under microscopy (**B**).

**Figure 2 jof-10-00243-f002:**
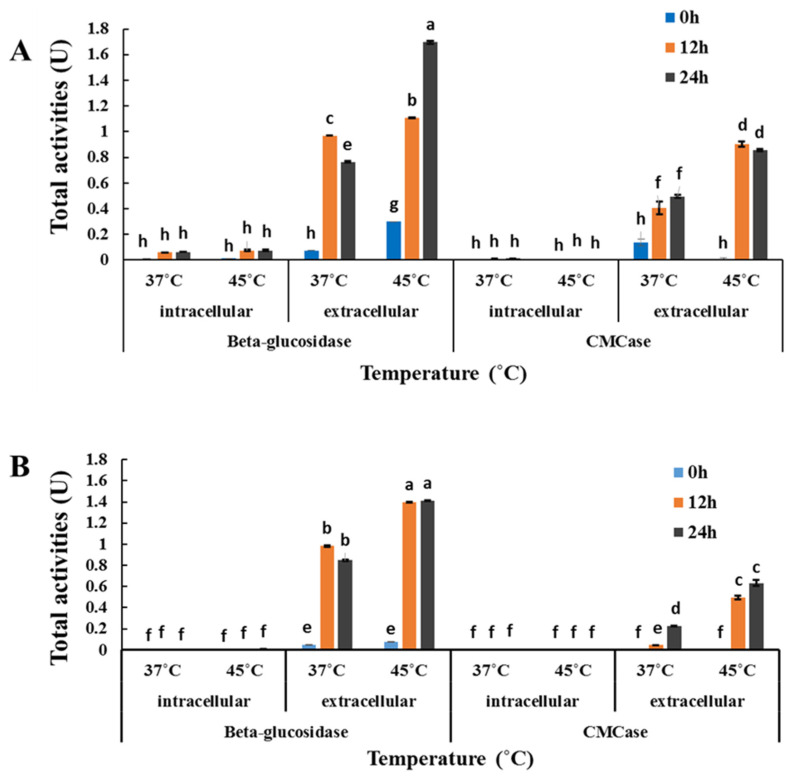
Enzyme activities of *C. rhodanensis* DK cultured at (**A**) 37 °C and (**B**) 45 °C. Data expressed as mean ± SD (N = 2). Different lowercase letters (a–h) indicate the significant differences of the values (*p* < 0.05).

**Figure 3 jof-10-00243-f003:**
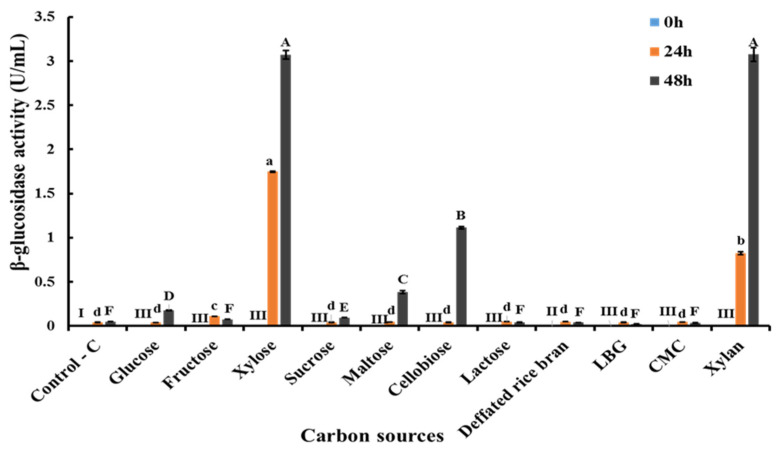
Extracellular β-glucosidase production by *C. rhodanensis* DK in yeast malt extract (YM) broth containing various carbon sources. Data expressed as mean ± SD (N = 2). Different Roman numeral (I, II, III) indicate the significant differences of the values (*p* < 0.05) at 0 h. Different lower letters (a–d) indicate the significant differences of the values (*p* < 0.05) at 24 h. Different upper letters (A–F) indicate the significant differences of the values (*p* < 0.05) at 48 h.

**Figure 4 jof-10-00243-f004:**
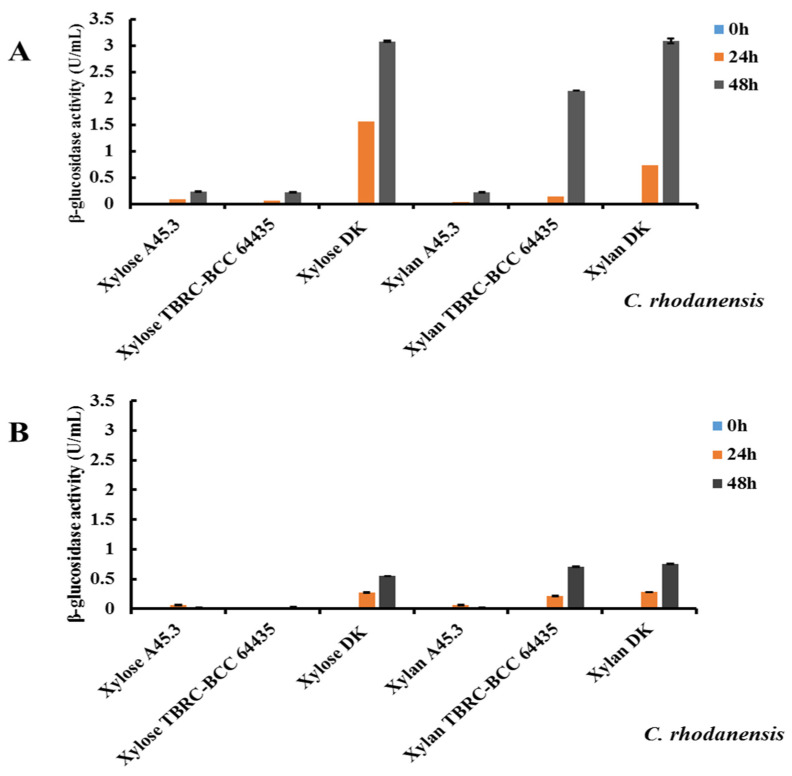
Extracellular β-glucosidase productions by *C. rhodanensis* (DK) and reference strains (TBRC-BCC 64435 and A45.3) incubation at (**A**) 37 °C and (**B**) 45 °C for 48 h accompanied by xylose and xylan. Data expressed as mean ± SD (N = 2).

**Figure 5 jof-10-00243-f005:**
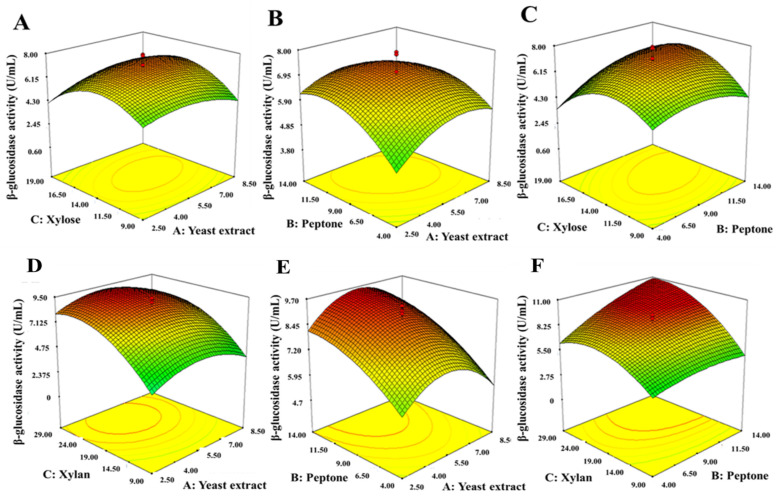
Three-dimensional response surface plots illustrate the interactions between variables and their influence on extracellular β-glucosidase production. (**A**–**C**) results are from xylose and (**D**–**F**) results are from xylan as carbon sources. (**A**) xylose and yeast extract, (**B**) peptone and yeast extract, (**C**) xylose and peptone, (**D**) xylan and yeast extract, (**E**) peptone and yeast extract, and (**F**) xylan and peptone. Different colors typically represent the dependent variable and red dots represent the actual data points used to create the smooth surface.

**Figure 6 jof-10-00243-f006:**
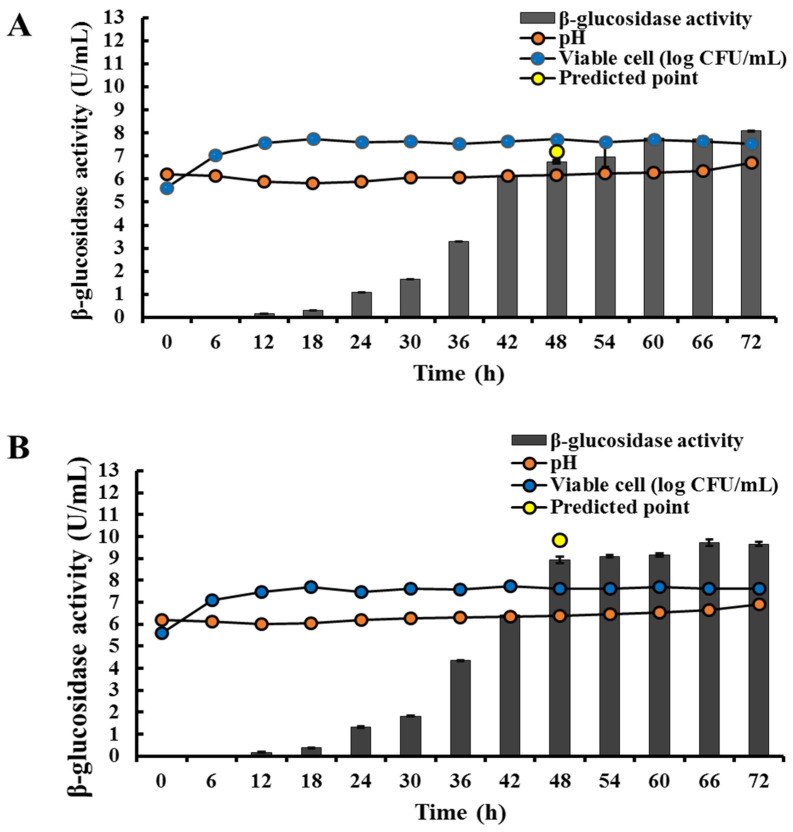
Validation of extracellular β-glucosidase production by *C. rhodanensis* DK using optimizaed medium. (**A**) Xylose and (**B**) xylan as the sole carbon sources.

**Figure 7 jof-10-00243-f007:**
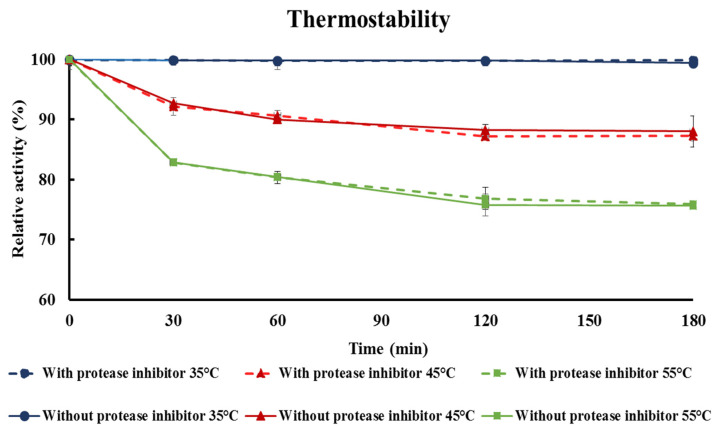
Time course of remaining relative activity (%) of crude β-glucosidase from *Cyberlindnera. rhodanensis* DK after incubated at 35, 45 and 55 °C.

**Table 1 jof-10-00243-t001:** Evaluation for the potential abilities of thermotolerant yeast isolates in the tolerance of tannin-tolerant ability and production of cellulases, β-mannanase, pectinase, xylanase, amylase and ethanol production cultured at 45 °C.

Isolates	Tannin- Tolerant Ability	Cellulases	β- Mannanase	Pectinase	Xylanase	Amylase	Ethanol Production
DK	++	+	−	−	−	−	+
MD1	++	+	−	−	−	−	−
MD2	++	+	−	−	−	−	−
MD3	+++	+	−	−	−	−	−
MD6	+++	+	−	−	−	−	−
MD7	++	+	−	−	−	−	−
MD8	++	+	−	−	−	−	−
MD12	+++	+	−	−	−	−	−
MD14	+++	+	−	−	−	−	−
MD20	+++	+	−	−	−	−	−
MD21	++	+	−	−	−	−	−
MD23	++	+	−	−	−	−	−
MD24	++	+	−	−	−	−	−
TN1	++	+	−	−	−	−	−
TN2	++	+	−	−	−	−	−
TN4	++	+	−	−	−	−	−
TN5	+++	+	−	−	−	−	−
TN7	++	+	−	−	−	−	−

(−): did not produce extracellular enzyme and ethanol; (+): produced extracellular enzymes and ethanol; (++): tolerant at a tannic acid concentration of 1–3% (*w*/*v*); (+++): tolerant at a tannic acid concentration of 5% (*w*/*v*).

**Table 2 jof-10-00243-t002:** Plackett–Burman experimental design with 15 runs and the responding β-glucosidase activity under the condition of using xylose and/or xylan as a substrate.

Run	A: Yeast Extract (g/L)	B: Peptone (g/L)	C: Malt Extract (g/L)	D: KH_2_PO_4_ (g/L)	E: MgSO_4_ (g/L)	F: Xylose or Xylan (g/L)	β-Glucosidase (Xylose) (U/mL)	β-Glucosidase (Xylan) (U/mL)
1	5.5	9	0.5	1.5	0.9	19	7.76	4.87
2	0.5	9	5.5	0.5	0.9	19	1.00	1.85
3	5.5	1	5.5	1.5	0.1	19	0.62	2.91
4	0.5	9	0.5	1.5	0.9	1	0.02	0.02
5	0.5	1	5.5	0.5	0.9	19	0.11	1.51
6	0.5	1	0.5	1.5	0.1	19	0.05	0.09
7	5.5	1	0.5	0.5	0.9	1	0.03	0.05
8	5.5	9	0.5	0.5	0.1	19	7.56	6.57
9	5.5	9	5.5	0.5	0.1	1	0.98	0.35
10	0.5	9	5.5	1.5	0.1	1	1.57	0.17
11	5.5	1	5.5	1.5	0.9	1	0.05	0.06
12	0.5	1	0.5	0.5	0.1	1	0.03	0.06
13	3	5	3	1	0.5	10	2.43	3.67
14	3	5	3	1	0.5	10	2.89	3.76
15	3	5	3	1	0.5	10	3.39	3.36

**Table 3 jof-10-00243-t003:** Statistical analysis of the Plackett–Burman design showing coefficient values and *p*-value for each variable under the condition of using xylose and/or xylan as a substrate.

Variable	Xylose		Xylan	
	Coefficient	*p*-Value	Coefficient	*p*-Value
Intercept	1.69	0.0308	1.54	0.0066
Yeast extract	1.14	0.0417	0.93	0.0095
Peptone	1.46	0.0155	0.70	0.0212
Malt extract	−0.97	0.0729	−0.40	0.01557
MgSO_4_	−0.19	0.6851	−0.19	0.5659
KH_2_PO_4_	0.071	0.8814	−0.15	0.4725
Xylose	1.24	0.0302	-	-
Xylan	-	-	1.42	0.0012
	R^2^ = 0.8016		R^2^ = 0.9188	

**Table 4 jof-10-00243-t004:** Using the CCD for optimization of β-glucosidase activity with xylose as a carbon source. CCD matrix of three variables in coded units and observed responses.

Run	A: Yeast Extract (g/L)	B: Peptone (g/L)	C: Xylose (g/L)	β-Glucosidase (U/mL)
1	2.5(−1)	4(−1)	9(−1)	4.44
2	8.5(+1)	4(−1)	9(−1)	4.83
3	2.5(−1)	14(+1)	9(−1)	4.95
4	8.5(+1)	14(+1)	9(−1)	4.64
5	2.5(−1)	4(−1)	29(+1)	1.32
6	8.5(+1)	4(−1)	29(+1)	4.01
7	2.5(−1)	14(+1)	29(+1)	5.06
8	8.5(+1)	14(+1)	29(+1)	6.04
9	0.45(−α)	9(0)	19(0)	4.69
10	10.55(+α)	9(0)	19(0)	4.17
11	5.5(0)	0.59(−α)	19(0)	3.86
12	5.5(0)	17.41(+α)	19(0)	4.52
13	5.5(0)	9(0)	2.18(−α)	0.62
14	5.5(0)	9(0)	35(+α)	2.90
15	5.5(0)	9(0)	19(0)	6.89
16	5.5(0)	9(0)	19(0)	6.25
17	5.5(0)	9(0)	19(0)	7.91
18	5.5(0)	9(0)	19(0)	7.82
19	5.5(0)	9(0)	19(0)	7.12
20	5.5(0)	9(0)	19(0)	6.79

The symbols and numbers (−α, −1, 0, +1, +α) represent the range and levels of the independent variables. Alpha (α) specifically denotes the axial points, which represent the lowest and highest coded values used in the design.

**Table 5 jof-10-00243-t005:** Using the CCD for optimization of β-glucosidase activity with xylan as a carbon source. CCD matrix of three variables in coded units and observed responses.

Run	A: Yeast Extract (g/L)	B: Peptone (g/L)	C: Xylan (g/L)	β-Glucosidase (U/mL)
1	2.5(−1)	4(−1)	9(−1)	3.30
2	8.5(+1)	4(−1)	9(−1)	3.24
3	2.5(−1)	14(+1)	9(−1)	4.11
4	8.5(+1)	14(+1)	9(−1)	4.23
5	2.5(−1)	4(−1)	29(+1)	4.99
6	8.5(+1)	4(−1)	29(+1)	4.11
7	2.5(−1)	14(+1)	29(+1)	9.45
8	8.5(+1)	14(+1)	29(+1)	9.34
9	0.45(−α)	9(0)	19(0)	4.92
10	10.55(+α)	9(0)	19(0)	4.76
11	5.5(0)	0.59(−α)	19(0)	5.53
12	5.5(0)	17.41(+α)	19(0)	8.94
13	5.5(0)	9(0)	2.18(−α)	0.07
14	5.5(0)	9(0)	35(+α)	7.99
15	5.5(0)	9(0)	19(0)	9.18
16	5.5(0)	9(0)	19(0)	8.29
17	5.5(0)	9(0)	19(0)	9.13
18	5.5(0)	9(0)	19(0)	8.31
19	5.5(0)	9(0)	19(0)	9.37
20	5.5(0)	9(0)	19(0)	8.98

The symbols and numbers (−α, −1, 0, +1, +α) represent the range and levels of the independent variables. Alpha (α) specifically denotes the axial points, which represent the lowest and highest coded values used in the design.

## Data Availability

Data are contained within the article.

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
