# Peer review of "A Thermotolerant Yeast Cyberlindnera rhodanensis DK Isolated from Laphet-so Capable of Extracellular Thermostable β-Glucosidase Production"

_jof, 2024, doi:10.3390/jof10040243_

Round 1

Reviewer 1 Report

Comments and Suggestions for Authors

The authors made a very strong research study. A vast variety of methodologies was used and all of them were very properly described in a scientific manner, rendering the processes clear and reproducible for the reader. The Results/Discussion was presented very explicitly; all the data were clearly presented and a significant number of literature studies were effectively used throughout the whole text to make many comments on the evaluation of the findings. Below, you will find some minor changes I propose prior to the publication of the study.

-Lines 90-91: I suggest you add more detailed advantages of microbial enzymes to plant/animal enzymes.

-Lines 419-420: In Figure 1, denote the letter (a) at the first image and the letter (b) at the second image. Also, be more descriptive of what each image represents in the figure legend.

Comments on the Quality of English Language

English language is fine. Minor revision can be made to avoid any unintentional mistakes.

Author Response

RESPONSE TO REVIEWER 1

Thank you for giving us the opportunity to revise the manuscript. We have revised/responded to each comment in a point-by-point pattern. All corrections were highlighted with the blue color in the revised manuscript (R1) and this document.

Response to Reviewer 1

General Comment: The authors made a very strong research study. A vast variety of methodologies was used and all of them were very properly described in a scientific manner, rendering the processes clear and reproducible for the reader. The Results/Discussion was presented very explicitly; all the data were clearly presented and a significant number of literature studies were effectively used throughout the whole text to make many comments on the evaluation of the findings. Below, you will find some minor changes I propose prior to the publication of the study.

General Response: We are grateful to the kind and positive comments on our manuscript.

Comment 1: Lines 90-91: I suggest you add more detailed advantages of microbial enzymes to plant/animal enzymes.

Response 1: Thank you for the suggestion. We have added the detailed advantages of microbial enzymes as suggested (Lines 93-101 in R1 version).

Comment 2: Lines 419-420: In Figure 1, denote the letter (a) at the first image and the letter (b) at the second image. Also, be more descriptive of what each image represents in the figure legend.

Response 2: The figure legend in Figure 1 was amended to denote the letter (A) for the first image and the letter (B) for the second image. Additionally, a more descriptive legend was provided (Lines 465).

Comment on the Quality of English Language: English language is fine. Minor revision can be made to avoid any unintentional mistakes.

Response: We appreciated your feedback. If there are more specific suggestions you needed to improve, we would be happy to revise following the reviewer’s guidance.

Reviewer 2 Report

Comments and Suggestions for Authors

This study aimed to isolate and explore the microbial species found within Laphet-so.  In the study, 18 thermotolerant yeasts were isolated that showed resistance to high concentrations of tannins, as well as the ability to degrade CMC through the production of CMCase. However, only the yeast identified as Cyberlindnera rhodanensis through sequencing studies of the D1/D2 domain in the rRNA gene showed the potential to produce ethanol at 45 ºC. Although the work is interesting and several results are shown, some observations need to be met in order for it to be publishable.

- The introduction needs to connect the importance between the existence of thermotolerant strains, tolerance to high concentrations of tannins and the production of B-glucosidase. It is not yet clear how important these characteristics are in a wild strain. The same should be applied when discussing the results. The authors do not provide a robust discussion of the data obtained. We have the impression that the article concentrates data from independent articles that have been relocated to form a single publication.

- What is the need to obtain a thermotolerant yeast that produces ethanol at 45ºC? Why was only this species considered a potential application in biorefineries? Aren't other species with the potential to grow in high tannin concentrations, as well as those that produce B-glucosidases, also interesting?

- In the Plackett-Burman design carried out, the concentration of malt agar had a negative effect compared to xylan as a carbon source. Wasn't an experiment carried out with only this carbon source in the medium to check the conditions?

- The conclusions should be more faithful to the results found and offer the authors' opinion on future studies.

Comments on the Quality of English Language

nothing to declare

Author Response

RESPONSE TO REVIEWER 2

Thank you for giving us the opportunity to revise the manuscript. We have considered each of the comments and have provided revisions/responses to each comment in a point-by-point pattern. All corrections by the authors are made with yellow color highlighted text in the revised manuscript (R1) and this document.

Response to Reviewer 2

General Comment: This study aimed to isolate and explore the microbial species found within Laphet-so.  In the study, 18 thermotolerant yeasts were isolated that showed resistance to high concentrations of tannins, as well as the ability to degrade CMC through the production of CMCase. However, only the yeast identified as Cyberlindnera rhodanensis through sequencing studies of the D1/D2 domain in the rRNA gene showed the potential to produce ethanol at 45ËšC. Although the work is interesting and several results are shown, some observations need to be met in order for it to be publishable.

General Response: We appreciate the reviewer thoughtful comments and recognition of this study significance.

Comment 1: The introduction needs to connect the importance between the existence of thermotolerant strains, tolerance to high concentrations of tannins and the production of β-glucosidase. It is not yet clear how important these characteristics are in a wild strain. The same should be applied when discussing the results. The authors do not provide a robust discussion of the data obtained. We have the impression that the article concentrates data from independent articles that have been relocated to form a single publication.

Response 1: Thank you for the valuable feedback. We acknowledge the need to strengthen the connection between the existence of thermotolerant strains, tolerance to high tannin concentrations, and β-glucosidase production in the introduction. We have revised to emphasize the significance of these characteristics in wild strains (Line 69-80).

We have also added a discussion to provide a more robust analysis of the obtained data (Line 330-333, 374-377, 602-603, 608-609).

Comment 2: What is the need to obtain a thermotolerant yeast that produces ethanol at 45ËšC? Why was only this species consider a potential application in biorefineries? Aren't other species with the potential to grow in high tannin concentrations, as well as those that produce β-glucosidase, also interesting?

Response 2: Biofuels or ethanol production has primarily relied on feedstocks rich in simple sugar such as glucose, sucrose, or molasses. However, advancements in technology have paved the way for more sustainable alternatives, particularly lignocellulosic biomasses, which have the potential to meet energy demands in an economically viable and environmentally safe manner without adversely affecting food supplies or the environment. However, additional processes to degrade the complexed lignocelluloses are required especially the saccharification process using external enzymes such as cellulases, hemicellulases, and beta-glucosidase to convert polysaccharides to monosaccharides which can be subsequently converted to ethanol by microbial metabolic pathway. Nonetheless, during the saccharification process, contamination by mesophilic microbes such as bacteria and wild yeast is a significant concern. To reduce or avoid contamination by those mesophilic microbes, thermotolerant yeast strains capable of fermenting at higher temperatures have been introduced.

The selected thermotolerant yeast species like the DK isolate is considered a promising yeast strain due to its ability in conversion of a cellulosic material representative, such as CMC to ethanol. The key feature of DK lies in its capability to produce CMCase at elevated temperatures, thereby facilitating effective cellulose hydrolysis and subsequent ethanol fermentation. This makes the yeast DK a suitable candidate for ethanol production under varying fermentation process conditions. Other characteristics, such as the capability to grow in the presence of tannin, are also of interest. However, further experiments are required to confirm the specific types of enzymes produced by this selected yeast species and to identify them for future studies. The choice between DK and these alternatives is based on a comprehensive evaluation of attributes such as ethanol production efficiency, tannin tolerance, and enzyme capabilities, ensuring a tailored fit for specific biorefinery processes.

Comment 3: In the Plackett-Burman design carried out, the concentration of malt agar had a negative effect compared to xylan as a carbon source. Wasn't an experiment carried out with only this carbon source in the medium to check the conditions?

Response 3: In the Plackett-Burman design conducted, we did not specifically carry out an experiment with only malt agar as the carbon source. Regarding the principle theory of the statistical design, the experiment focused on screening various nutritional factors, including different concentrations of multiple factors, to identify both their individual and interaction impacts on the response or outcome of interest, and the most 2 or 3 influence nutritional factors will be selected for optimization to achieve the metabolic products (b-glucosidase in this research).  According to the theory, the observed negative effect of yeast malt extract medium or malt agar in the presence of xylan indicated that malt agar was not a favorable carbon source compared to xylan. However, neither malt agar nor xylan alone demonstrated a significant influence on enzyme production. In our case, it is possible that the enzyme induction by xylan and malt extract agar had a negative interaction, and our selected yeast strain prefers to utilize xylan as the carbon source compared to malt extract agar.

Comment 4: The conclusions should be more faithful to the results found and offer the authors' opinion on future studies.

Response 4: We appreciated this feedback and acknowledged the importance of aligning our conclusions closely with the findings. The conclusion was revised and the perspective on potential avenues for future research has been added. (Line 854-867).

Reviewer 3 Report

Comments and Suggestions for Authors

A Thermotolerant Yeast Cyberlindnera rhodanensis DK Isolated from Laphet-so Capable of Extracellular Thermostable β-Glucosidase Production

Nang Nwet Noon Kham, Somsay Phovisay, Kridsada Unban, Apinun Kanpiengjai, Chalermpong Saenjum, Saisamorn Lumyong, Kalidas Shetty and Chartchai Khanongnuch

This research covers many tasks associated with applied microbiology, molecular biology, and enzymology. I suppose, that studies are innovative and have a practical value. This study had the goal to apply the microbial resources found from Laphet-so. A total of eighteen isolates of thermotolerant yeasts were obtained from eight samples of Laphet-so. It is a valuable to mention that in this study all isolates demonstrated the tannin tolerance and six isolates were resistant to 5% (w/v) tannin concentration. Also, eighteen isolates were capable of carboxy-methyl cellulose (CMC) degrading. This research makes the sense for the future studies.

Some comments and questions to the authors

1)     Could you describe in more detail practical/industrial value in microbiology of some Cyberlindnera rhodanensis strains in the introduction part?

2)     In the Figure 3 on the x axis some carbon sources are written in lowercase letter, sometimes in uppercase letter for example „glucose, Fructose“. Please, make it the same.

3)     Which strains did you describe of thermotolerant yeast isolates in the Table 1?

4)     How the authors could explain why using xylose and xylan had the most activity of β-glucosidase?

5)     Which renewable carbon source or sources the authors could suggest for the future studies for the extraction of β-glucosidase?

6)     Which other polysaccharides the authors could suggest for the future studies?

7)     Which high tannin containing substrates the authors could suggest for food biotechnology?

Author Response

RESPONSE TO REVIEWER 3

Thank you for giving us the opportunity to revise the manuscript. We have revised our manuscript considered each of the comments and have provided revisions/responses to each comment in a point-by-point pattern. All corrections are made with green color highlighted text in the revised manuscript (R1) and this document.

Response to Reviewer 3

General Comment: This research covers many tasks associated with applied microbiology, molecular biology, and enzymology. I suppose, that studies are innovative and have a practical value. This study had the goal to apply the microbial resources found from Laphet-so. A total of eighteen isolates of thermotolerant yeasts were obtained from eight samples of Laphet-so. It is a valuable to mention that in this study all isolates demonstrated the tannin tolerance and six isolates were resistant to 5% (w/v) tannin concentration. Also, eighteen isolates were capable of carboxy-methyl cellulose (CMC) degrading. This research makes the sense for the future studies.

General Response: We appreciate the reviewer’s recognition of its potential implications for further investigations.

Comment 1: Could you describe in more detail practical/industrial value in microbiology of some Cyberlindnera rhodanensis strains in the introduction part?

Response 1: We have elaborated further on the practical and industrial significance of C. rhodanensis strains in microbiology in the revised introduction. This expansion is highlighted in green color in the introduction to be significance. (Lines 120-126)

Comment 2: In the Figure 3 on the x axis some carbon sources are written in lowercase letter, sometimes in uppercase letter for example “glucose, Fructose”. Please, make it the same.

Response 2: Thank you for bringing this to my attention. We have revised Figure 3 and made the necessary adjustments to ensure uniformity in the capitalization of carbon sources on the X-axis. (Lines 594-595)

Comment 3: Which strains did you describe of thermotolerant yeast isolates in the Table 1?

Response 3: All yeast isolates presented in Table 1 are characterized as thermotolerant strains. The table comprehensively described details of the strains, their respective properties, and relevant characteristics relating to thermotolerance.

Comment 4: How the authors could explain why using xylose and xylan had the most activity of β-glucosidase?

Response 4: Most of β-glucosidase activity observed with xylose and xylan can be attributed to several factors. Firstly, these carbon sources in the experimental setup may act as inducers, potentially up-regulating the expression of gene encoding β-glucosidase in C. rhodanensis DK DNA and thereby enhancing its overall activity. The evidence confirms the induction of b-glucosidase production by either xylose or xylan, which was also previously reported in the case of β -glucosidase production by Aspergillus saccharolyticus, as described by Sørensen et al. (2014). Additionally, xylose can stimulate or maintain the activity of β -glucosidase from the thermophilic fungus Humicola insolens (Souza et al., 2013). This increased enzyme catalytic efficiency may also be attributed to the structural compatibility between the active site of β-glucosidase and the xylose and xylan molecules, facilitating more effective substrate binding and subsequent catalysis.

Souza et al. (2013) Glucose and xylose stimulation of a β-glucosidase from the thermophilic fungus Humicola insolens: A kinetic and biophysical study, Journal of Molecular Catalysis B,  94:119-128

Comment 5: Which renewable carbon source or sources the authors could suggest for the future studies for the extraction of β-glucosidase?

Response 5: According to our results, the enzyme production are produced in the highest level, with non-significant difference when xylose and xylan were used as inducers.  Xylose is considering to be used as a carbon source for future studies on β-glucosidase extraction due to its availability and the lower cost compared to xylan and cellobiose. This makes xylose a practical and economical choice for β-glucosidase extraction. However, focusing on renewable carbon source properties makes a reasonable consideration on the xylan containing agricultural wastes/residues such as corncob or corn stover, sugar cane bagasse, wheat bran, rice bran, etc.

Comment 6: Which other polysaccharides the authors could suggest for the future studies?

Response 6: Extending from response No. 5, future studies should be tried on some agricultural residues such as corn stover, wheat straw, and sugarcane bagasse as these materials are known to contain substantial amounts of cellulose, cellobiose, and also xylan. Some agricultural residues mentioned have been reported for the abundance of xylan, an effective β-glucosidase inducer confirmed by our result in this study.

Comment 7: Which high tannin containing substrates the authors could suggest for food biotechnology?

Response 7: The high tannins containing substrates for food biotechnology include tea leaves, coffee byproducts, hops, grape pomace, chestnut tannins, tannin-rich forages, etc. These substrates are rich in polyphenolic compounds like tannins or tannin-related compounds, which offer antioxidant and antimicrobial activities. Besides, those compounds are also responsible for flavor and aroma enhancement, extending shelf life, and improving the overall quality of food products.